# PODS: POLICY OPTIMIZATION VIA DIFFERENTIABLE SIMULATION

## ABSTRACT

Current reinforcement learning (RL) methods use simulation models as simple black-box oracles. In this paper, with the goal of improving the performance exhibited by RL algorithms, we explore a systematic way of leveraging the additional information provided by an emerging class of differentiable simulators. Building on concepts established by Deterministic Policy Gradients (DPG) methods, the neural network policies learned with our approach represent deterministic actions. In a departure from standard methodologies, however, learning these policy does not hinge on approximations of the value function that must be learned concurrently in an actor-critic fashion. Instead, we exploit differentiable simulators to directly compute the analytic gradient of a policy's value function with respect to the actions it outputs. This, in turn, allows us to efficiently perform locally optimal policy improvement iterations. Compared against other state-of-the-art RL methods, we show that with minimal hyper-parameter tuning our approach consistently leads to better asymptotic behavior across a set of payload manipulation tasks that demand a high degree of accuracy and precision.

## 1 INTRODUCTION

The main goal in RL is to formalize principled algorithmic approaches to solving sequential decision-making problems. As a defining characteristic of RL methodologies, agents gain experience by acting in their environments in order to learn how to achieve specific goals. While learning directly in the real world (Haarnoja et al., 2019; Kalashnikov et al., 2018) is perhaps the holy grail in the field, this remains a fundamental challenge: RL is notoriously data hungry, and gathering real-world experience is slow, tedious and potentially unsafe. Fortunately, recent years have seen exciting progress in simulation technologies that create realistic virtual training grounds, and sim-2-real efforts (Tan et al., 2018; Hwangbo et al., 2019) are beginning to produce impressive results.

A new class of *differentiable* simulators (Zimmermann et al., 2019; Liang et al., 2019; de Avila Belbute-Peres et al., 2018; Degrave et al., 2019) is currently emerging. These simulators not only predict the outcome of a particular action, but they also provide derivatives that capture the way in which the outcome will change due to infinitesimal changes in the action. Rather than using simulators as simple black box oracles, we therefore ask the following question: how can the additional information provided by differentiable simulators be exploited to improve RL algorithms?

To provide an answer to this question, we propose a novel method to efficiently learn control policies for finite horizon problems. The policies learned with our approach use neural networks to model deterministic actions. In a departure from established methodologies, learning these policies does not hinge on learned approximations of the system dynamics or of the value function. Instead, we leverage differentiable simulators to directly compute the analytic gradient of a policy's value function with respect to the actions it outputs for a specific set of points sampled in state space. We show how to use this gradient information to compute first and second order update rules for locally optimal policy improvement iterations. Through a simple line search procedure, the process of updating a policy avoids instabilities and guarantees monotonic improvement of its value function.

To evaluate the policy optimization scheme that we propose, we apply it to a set of control problems that require payloads to be manipulated via stiff or elastic cables. We have chosen to focus our attention on this class of high-precision dynamic manipulation tasks for the following reasons:

- they are inspired by real-world applications ranging from cable-driven parallel robots and crane systems to UAV-based transportation to (Figure 1);

- the systems we need to learn control policies for exhibit rich, highly non-linear dynamics;

- the specific tasks we consider constitute a challenging benchmark because they require very precise sequences of actions. This is a feature that RL algorithms often struggle with, as the control policies they learn work well on average but tend to output noisy actions. Given that sub-optimal control signals can lead to significant oscillations in the motion of the payload, these manipulation tasks therefore make it possible to provide an easy-to-interpret comparison of the quality of the policies generated with different approaches;

- by varying the configuration of the payloads and actuation setups, we can finely control the complexity of the problem to test systematically the way in which our method scales.

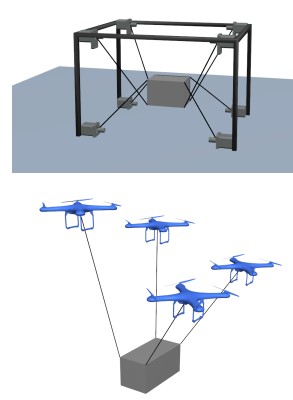

Figure 1: Real-world applications that inspire the control problems we focus on in this paper

The results of our experiments confirm our theoretical derivations and show that our method consistently outperforms two state-of-the-art (SOTA) model-free RL algorithms, Proximal Policy Optimization(PPO) (Wang et al., 2019) and Soft Actor-Critic(SAC) (Haarnoja et al., 2018), as well as the model-based approach of Backpropagation Through Time (BPTT). Although our policy optimization scheme (PODS) can be interleaved within the algorithmic framework of most RL methods (e.g. by periodically updating the means of the probability distributions represented by stochastic policies), we focused our efforts on evaluating it in isolation to pinpoint the benefits it brings. This allowed us to show that with minimal hyper-parameter tuning, the second order update rule that we derive provides an excellent balance between rapid, reliable convergence and computational complexity. In conjunction with the continued evolution of accurate differentiable simulators, our method promises to significantly improve the process of learning control policies using RL.

## 2 RELATED WORK

**Deep Reinforcement Learning.** Deep RL (DRL) algorithms have been increasingly more successful in tackling challenging continuous control problems in robotics (Kober et al., 2013; Li, 2018). Recent notable advances include applications in robotic locomotion (Tan et al., 2018; Haarnoja et al., 2019), manipulation (OpenAI et al., 2018; Zhu et al., 2019; Kalashnikov et al., 2018; Gu et al., 2016), and navigation (Anderson et al., 2018; Kempka et al., 2016; Mirowski et al., 2016) to mention a few. Many model-free DRL algorithms have been proposed over the years, which can be roughly divided into two classes, off-policy methods (Mnih et al., 2016; Lillicrap et al., 2016; Fujimoto et al., 2018; Haarnoja et al., 2018) and on-policy methods (Schulman et al., 2015; 2016; Wang et al., 2019), based on whether the algorithm can learn independently from how the samples were generated. Recently, model-based RL algorithms (Nagabandi et al., 2017; Kurutach et al., 2018; Clavera et al., 2018; Nagabandi et al., 2019) have emerged as a promising alternative for improving the sample efficiency. Our method can be considered as an on-policy algorithm as it computes first or second-order policy improvements given the current policy's experience.

**Policy Update as Supervised Learning.** Although policy gradient methods are some of the most popular approaches for optimizing a policy (Kurutach et al., 2018; Wang et al., 2019), many DRL algorithms also update the policy in a supervised learning (SL) fashion by explicitly aiming to mimic expert demonstration (Ross et al., 2011) or optimal trajectories (Levine & Koltun, 2013a;b; Mordatch & Todorov, 2015). Optimal trajectories, in particular, can be computed using numerical methods such as iterative linear–quadratic regulators (Levine & Koltun, 2013a;b) or contact invariant optimization (Mordatch & Todorov, 2015). The solutions they provide have the potential to improve the sample efficiency of RL methods either by guiding the learning process through meaningful samples (Levine & Koltun, 2013a) or by explicitly matching action distributions (Mordatch & Todorov, 2015). Importantly, these approaches are not only evaluated in simulation but have also been shown

to be effective for many real-world robotic platforms, including manipulators (Schenck & Fox, 2016; Levine et al., 2016) and exoskeletons (Duburcq et al., 2019). Recently, Peng et al. (2019) proposed an off-policy RL algorithm that uses SL both to learn the value function and to fit the policy to the advantage-weighted target actions. While our method shares some similarities with this class of approaches that interleave SL and RL, the updates of our policy do not rely on optimal trajectories that must be given as input. Rather, we show how to leverage differentiable simulators to compute locally optimal updates to a policy. These updates are computed by explicitly taking the gradient of the value function with respect to the actions output by the policy. As such, our method also serves to reinforce the bridge between the fields of trajectory optimization and reinforcement learning.

**Differentiable Models.** Our approach does not aim to learn a model of the system dynamics, but rather leverages differentiable simulators that explicitly provide gradients of simulation outcomes with respect to control actions. We note that traditional physics simulators such as ODE Drumwright et al. (2010) or PyBullet Coumans & Bai (2016–2019) are not designed to provide this information. We build, in particular, on a recent class of analytically differentiable simulators that have been shown to effectively solve trajectory optimization problems, with a focus on sim-2-real transfer, for both manipulation (Zimmermann et al., 2019) and locomotion tasks (Bern et al., 2019).

Degrave et al. (2019) embed a differentiable rigid body simulator within a recurrent neural network to concurrently perform simulation steps while learning policies that minimize a loss corresponding to the control objective. While their goal is related to ours, we show how to leverage explicitly-computed gradients to formulate second order policy updates that have a significant positive effect on convergence. Furthermore, in contrast to Degrave et al. (2019), we show that PODS consistently outperforms two common RL baselines, PPO (Wang et al., 2019) and SAC (Haarnoja et al., 2018).

Also related to our method is the very recent work of Clavera et al. (2020). Their observation is that while most model-based RL algorithms use models simply as a source of data augmentation or as a black-box oracle to sample from (Nagabandi et al., 2017), the differentiability of learned dynamics models can and should be exploited further. In an approach that is related to ours, they propose a policy optimization algorithm based on derivatives of the learned model. In contrast, we directly use differentiable simulators for policy optimization, bypassing altogether the need to learn the dynamics – including all the hyperparameters that are involved in the process, as well as the additional strategies required to account for the inaccuracies introduced by the learned dynamics (Boney et al., 2019). Thanks to the second order update rule that we derive, our method consistently outperforms SOTA model-free RL algorithms in the tasks we proposed. In contrast, their method only matches the asymptotic performance of model-free RL (which is a feat for model-based RL). It is also worth pointing out that while model-based approaches hold the promise of enabling learning directly in the real world, with continued progress in sim-2-real transfer, methods such as ours that rely on accurate simulation technologies will continue to be indispensable in the field of RL.

A common approach to leverage differentable models is that of backpropagating through time (BPTT) as is the main focus of Grzeszczuk et al. (1998), Deisenroth & Rasmussen (2011), Parmas (2018), Degrave et al. (2019), and Clavera et al. (2020), where a policy $\pi_\theta$ parametrized by $\theta$ is optimized directly in parameter space (PS), coupling the actions at each time step by the policy parameters. In contrast, our approach alternates between optimizing in trajectory space (TS), following gradient information of the value function for an independent set of actions $a_t = \pi_\theta(s)|_{s=s_t}$, and in parameter space (PS) by doing imitation learning of the monotonically improved actions $a_t$ by $\pi_\theta$. Alternating between TS and PS allows PODS to avoid the well-know problems of BPTT (vanishing and exploding gradients), that have been reported for a long time (Bengio et al., 1994).

## 3 POLICY OPTIMIZATION ON DIFFERENTIABLE SIMULATORS

Following the formulation employed by DPG methods, for a deterministic neural network policy $\pi_\theta$ parameterized by weights $\theta$, the RL objective $J(\pi_\theta)$ and its gradient $\nabla_\theta J(\pi_\theta)$ are defined as:

$$J(\pi_\theta) \quad = \int_S p(s_0) V^{\pi_\theta}(s_0) ds_0, \tag{1}$$

$$\nabla_\theta J(\pi_\theta) = \int_S p(s_0) \nabla_\theta V^{\pi_\theta}(s_0) ds_0 \ \approx \ \frac{1}{k} \sum_i^k \nabla_\theta V^{\pi_\theta}(s_{0,i}). \tag{2}$$

where $p(s_0)$ is the initial probability distribution over states, $V^{\pi_\theta}$ is the value function for $\pi_\theta$, and the second expression in Eq. 2 approximates the integral with a sum over a batch of $k$ initial states sampled from $S$, as is standard.

Restricting our attention to an episodic problem setup with fixed time horizon $N$ and deterministic state dynamics $s_{t+1} = f(s_t, a_t)$, the value function gradient simplifies to:

$$\nabla_{\boldsymbol{\theta}} V^{\pi_\theta}(s_0) = \nabla_{\boldsymbol{\theta}} \left( r(s_0, \pi_{\boldsymbol{\theta}}(s_0)) + \sum_{t=1}^{N} r(s_t, \pi_{\boldsymbol{\theta}}(s_t)) \right). \tag{3}$$

Noting that the state $s_t$ can be specified as a recursive function $s_t = f(s_{t-1}, \pi_{\boldsymbol{\theta}}(s_{t-1}))$, the computation of the gradient in Eq 3 is equivalent to backpropagating through time (BPTT) into the policy parameters. However, BPTT can be challenging due to well known problems of vanishing or exploding gradients (Degrave et al., 2019). We therefore turn our focus to the task of performing policy improvement iterations. In particular, our goal is to find a new policy $\bar{a}$, in trajectory space, such that $V^{\pi_\theta}(s_0) < V^{\bar{a}}(s_0)$ for a batch of initial states sampled according to $s_0 \sim p(s_0)$.

### 3.1 FIRST ORDER POLICY IMPROVEMENT

While the parametrization of $\pi_\theta$ is given in terms of $\theta$ (the weights of the neural net), we will choose TS policy $\bar{a}$ to directly have as parameters the actions that are executed at each time step. By representing the actions independently of each other, rather than having them coupled through $\theta$, BPTT is therefore not required. Moreover, at the start of each policy improvement step, we initialize the TS policy $\bar{a} = [a_0, a_1, \ldots, a_{N-1}]$ to match the output of $\pi_\theta$, where the individual terms $a_t$ are the actions executed during a rollout of $\pi_\theta(s)|_{s=s_{t-1}}$. Thus, $V^{\pi_\theta}(s_0) = V^{\bar{a}}(s_0)$ initially. The value function gradient of policy $\bar{a}$ is then:

$$\nabla_{\bar{a}} V^{\bar{a}}(s_0) = \nabla_{\bar{a}} V^{\bar{a}}(\boldsymbol{s}(\bar{a}), \bar{a}) = \nabla_{\bar{a}} \left( r(s_0, a_0) + \sum_{t=1}^{N} r(s_t(a_{t-1}), a_t) \right). \tag{4}$$

where $\boldsymbol{s}(\bar{a}) = [s_0, s_1(a_0), \ldots, s_N(a_{N-1})]$ is the vector of the state trajectory associated to the policy rollout. For the sake of clarity we now switch notation from $\nabla_{\bar{a}}$ to $\frac{\mathrm{d}(.)}{\mathrm{d}\bar{a}}$:

$$\frac{\mathrm{d}V^{\bar{a}}(s_0)}{\mathrm{d}\bar{a}} = \frac{\partial V^{\bar{a}}}{\partial \bar{a}} + \frac{\partial V^{\bar{a}}}{\partial \boldsymbol{s}} \frac{\mathrm{d}\boldsymbol{s}}{\mathrm{d}\bar{a}}. \tag{5}$$

For a known, differentiable reward, the terms $\frac{\partial V^{\bar{a}}}{\partial \bar{a}}$ and $\frac{\partial V^{\bar{a}}}{\partial \boldsymbol{s}}$ can be easily computed analytically. In contrast, the Jacobian $\frac{\mathrm{d}\boldsymbol{s}}{\mathrm{d}\bar{a}}$, that represents the way in which the state trajectory changes as the policy $\bar{a}$ changes, is the first piece of information that we will require from a differentiable simulator. Furthermore, notice that even though we are not BPTT, the lower triangular structure of $\frac{\mathrm{d}\boldsymbol{s}}{\mathrm{d}\bar{a}}$ encodes the dependency of a particular point in state space on all the previous actions during a rollout (see the Appendix A.5 for more details on the Jacobian structure.

The first order update rule for policy $\bar{a}$ is then computed as:

$$\bar{a} = \boldsymbol{\pi_\theta} + \alpha_a \frac{\mathrm{d}V^{\bar{a}}(s_0)}{\mathrm{d}\bar{a}}. \tag{6}$$

Since this update rule uses the policy gradient (i.e. the direction of local steepest ascent), there exists a value $\alpha_a > 0$ such that $V^{\pi_\theta}(s_0) < V^{\bar{a}}(s_0)$. In practice, we use the simulator to run a standard line-search on $\alpha_a$ to ensure the inequality holds. We note, however, that if desired, $\alpha_a$ can also be treated as a hyperparameter that is tuned to a sufficiently small value.

Once the policy $\bar{a}$ has been improved, we can use the corresponding state trajectories $\boldsymbol{s}(\bar{a})$ to update the parameters of the neural net policy $\pi_\theta$ by running gradient descent on the following loss:

$$L_{\boldsymbol{\theta}} = \frac{1}{k} \sum_{i}^{k} \sum_{t}^{N} \frac{1}{2} \|\pi_{\boldsymbol{\theta}}(s_{t,i}) - a_{t,i}\|^2, \tag{7}$$

where the gradient and update rule are given by:

$$\nabla_{\boldsymbol{\theta}} L_{\boldsymbol{\theta}} = \frac{1}{k} \sum_i^k \sum_t^N \nabla_{\boldsymbol{\theta}} \pi_{\boldsymbol{\theta}}(s_i)(\pi_{\boldsymbol{\theta}}(s_{t,i}) - a_{t,i}), \tag{8}$$

$$\boldsymbol{\theta} = \boldsymbol{\theta} - \alpha_{\boldsymbol{\theta}} \nabla_{\boldsymbol{\theta}} L_{\boldsymbol{\theta}}. \tag{9}$$

Here, $i$ indexes the batch of initial states used to approximate the integral in Eq 2. Notice that gradients $\nabla_{\boldsymbol{\theta}} J(\pi_{\boldsymbol{\theta}})$ and $\nabla_{\boldsymbol{\theta}} L_{\boldsymbol{\theta}}$ are closely related for the first iteration in the policy improvement operation, where:

$$\nabla_{\boldsymbol{\theta}} L_{\boldsymbol{\theta}} = -\alpha_{\boldsymbol{\theta}} \alpha_a \frac{1}{k} \sum_i^k \nabla_{\boldsymbol{\theta}} \boldsymbol{\pi}_{\boldsymbol{\theta}}(s_{0,i}) \frac{\mathrm{d} V^{\bar{a}}(s_{0,i})}{\mathrm{d} \bar{a}}, \tag{10}$$

which explains why minimizing Eq.7 improves the value function formulated in Eq. 1. It is also worth noting that the stability of the policy improvement process is guaranteed by the parameter $\alpha_a$, which is found through a line search procedure such that $V^{\pi_{\boldsymbol{\theta}}}(s_0) < V^{\bar{a}}(s_0)$, as well as through the intermediate targets of Eq. 7, which eliminate potential overshooting problems that might occur if the gradient direction in Eq.10 was followed too aggressively.

## 3.2 SECOND ORDER POLICY IMPROVEMENT

For a second order policy update rule, the Hessian $\frac{\mathrm{d}^2 V^{\bar{a}}(s_0)}{\mathrm{d}\bar{a}^2}$ is required. A brief derivation of this expression can be found in the Appendix and is summarized as follows:

$$\frac{\mathrm{d}^2 V^{\bar{a}}(s_0)}{\mathrm{d}\bar{a}^2} = \frac{\mathrm{d}}{\mathrm{d}\bar{a}} \left[ \frac{\partial V^{\bar{a}}}{\partial \bar{a}} + \frac{\partial V^{\bar{a}}}{\partial s} \frac{\mathrm{d}s}{\mathrm{d}\bar{a}} \right], \tag{11}$$

$$= \frac{\partial V^{\bar{a}}}{\partial s} \left( \frac{\mathrm{d}s}{\mathrm{d}\bar{a}}^T \frac{\partial}{\partial s} \frac{\mathrm{d}s}{\mathrm{d}\bar{a}} + \frac{\partial}{\partial \bar{a}} \frac{\mathrm{d}s}{\mathrm{d}\bar{a}} \right) + \frac{\mathrm{d}s}{\mathrm{d}\bar{a}}^T \left( \frac{\partial^2 V^{\bar{a}}}{\partial s^2} \frac{\mathrm{d}s}{\mathrm{d}\bar{a}} + 2 \frac{\partial^2 V^{\bar{a}}}{\partial s \partial \bar{a}} \right) + \frac{\partial^2 V^{\bar{a}}}{\partial \bar{a}^2}. \tag{12}$$

The second order tensors $\frac{\partial}{\partial s} \frac{\mathrm{d}s}{\mathrm{d}\bar{a}}$ and $\frac{\partial}{\partial \bar{a}} \frac{\mathrm{d}s}{\mathrm{d}\bar{a}}$ are additional terms that a differentiable simulator must provide. As described in Zimmermann et al. (2019), these terms can be computed analytically. However, they are computationally expensive to compute, and they often lead to the Hessian becoming indefinite. As a consequence, ignoring these terms from the equation above results in a Gauss-Newton approximation of the Hessian:

$$\frac{\mathrm{d}^2 V^{\bar{a}}(s_0)}{\mathrm{d}\bar{a}^2} \approx \hat{\mathbf{H}} = \frac{\mathrm{d}s}{\mathrm{d}\bar{a}}^T \frac{\partial^2 V^{\bar{a}}}{\partial s^2} \frac{\mathrm{d}s}{\mathrm{d}\bar{a}} + \frac{\partial^2 V^{\bar{a}}}{\partial a^2}. \tag{13}$$

In the expression above we assume that the rewards do not couple $s$ and $a$. As long as the second derivatives of the rewards with respect to states and actions are positive definite, which is almost always the case, the Gauss-Newton approximation $\hat{\mathbf{H}}$ is also guaranteed to be positive semi-definite. A second order update rule for $\bar{a}$ can therefore be computed as:

$$\bar{a} = \boldsymbol{\pi}_{\boldsymbol{\theta}} + \alpha_a \hat{\mathbf{H}}^{-1} \frac{\mathrm{d} V^{\bar{a}}(s_0)}{\mathrm{d}\bar{a}}. \tag{14}$$

Analogous to the first order improvements discussed in the previous section, the same loss $L_{\boldsymbol{\theta}}$ can be used to perform a policy update on $\pi_{\boldsymbol{\theta}}$ to strictly improve its value function. In this case, $L_{\boldsymbol{\theta}}$ incorporates the second order policy updates of Eq. 14 without the need to compute the Hessian of the neural network policy, and with the additional benefit of allowing the use of well-defined acceleration methods such as Adam (Kingma & Ba, 2015).

## 3.3 MONOTONIC POLICY IMPROVEMENT

The combination of a simple line search on $\alpha_a$ together with the use of $L_{\boldsymbol{\theta}}$ to update $\pi_{\boldsymbol{\theta}}$ provides a simple and very effective way of preventing overshooting as $\boldsymbol{\theta}$ is updated. PODS therefore features

monotonic increases in performance, as shown through our experiments. As summarized in Figure 2 for the task of controlling a 2D pendulum such that it goes to stop as quickly as possible (see the experiments section for a detailed description of task), both the first and second order policy improvement methods are well-behaved. Nevertheless, there is a drastic difference in convergence rates, with the second order method winning by a significant margin.

---

**Algorithm 1:** PODS: Policy Optimization via Differentiable Simulators

---

**for** *epoch = 1, M* **do**
    **for** *sample i = 1, k* **do**
        Sample initial condition $s_{0,i}$
        Collect $\pi_{\theta}$ by rolling out $\pi_{\theta}$ starting from $s_{0,i}$
        Compute improved policy $\bar{a}_i$ (Eq 6. or Eq 14.)
    **end**
    Run gradient descent on $L_{\theta}$ (Eq 7.) such that the output
    of $\pi_{\theta}$ matches $\bar{a}_i$ for the entire sequence of states $\mathbf{s}(\bar{a}_i)$
**end**

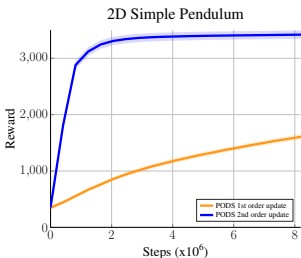

Figure 2: Performance of first and second order update rules.

In contrast to other approaches such as PPO (Wang et al., 2019) and SAC (Haarnoja et al., 2018), our policy update scheme does not need to be regularized by a KL-divergence metric, demonstrating its numerical robustness. Our method is only limited by the expressive power of policy $\pi_{\theta}$, as it needs to approximate $\bar{a}$ well. For reasonable network architectures, this is not a problem, especially since $\bar{a}$ corresponds to local improvements. The overall PODS algorithm is summarized above. For the experiments we present in the next section, we collected $k = 4000$ rollouts for each epoch, and we performed 50 gradient descent steps on $L_{\theta}$ for each policy optimization iteration.

## 4 EXPERIMENTS

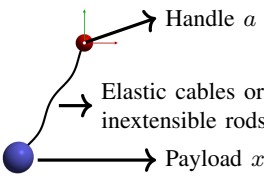
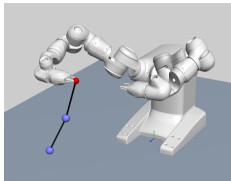
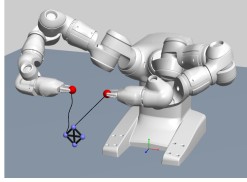
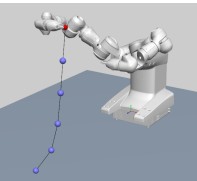

Figure 3: Experiments left to right; 2D pendulum, 3D double pendulum, Cable driven payload 2D, Discretized 3D rope

**Environments:** The environments used in our experiments set up cable-driven payload manipulation control problems that are inspired by the types of applications visualized in Figure 1. For all these examples, as illustrated in Figure 3, the action space is defined by the velocity of one or more *handles*, which are assumed to be directly controlled by a robot, and the state space is defined by the position of the handle as well as the position and velocity of the *payload*. We model our dynamical systems as mass-spring networks by connecting payloads to handles or to each other via stiff bilateral or unilateral springs. Using a simulation engine that follows closely the description in Zimmermann et al. (2019), we use a BDF2 integration scheme, as it exhibits very little numerical damping and is stable even under large time steps. Although this is not a common choice for RL environments, the use of higher order integration schemes also improves simulation quality and accuracy, as pointed out by Zhong et al. (2020). The Jacobian $\frac{\mathrm{d}\boldsymbol{s}}{\mathrm{d}\bar{\boldsymbol{a}}}$, which is used for both the first order and second order policy updates, is computed analytically via sensitivity analysis, as described in detail Zimmermann et al. (2018). The computational cost of computing this Jacobian is significantly less than performing the sequence of simulation steps needed for a policy rollout.

The control problems we study here are deceptively simple. All the environments fall in the category of underactuated systems and, in consequence, policies for such environments must fully leverage the system's dynamics to successfully achieve a task. The lack of numerical damping in the motion's payload, in particular, necessitates control policies that are very precise, as even small errors lead to

noticeable oscillations. These environments also enable us to incrementally increase the complexity of the tasks in order to study the scalability of our method, as well as that of the RL algorithms we compare against. For comparison purposes, in particular, we use three different types of dynamical systems; 2D Simple Pendulum, 3D Simple Pendulum, and 3D Double Pendulum. A detailed description of these environments is presented in Appendix A.2.

For all the environments, the action space describes instantaneous velocities of the handles, which are restricted to remain within physically reasonable limits.

**Tasks:**  In order to encode our tasks, we used continuous rewards that are a function of the following state variables: the position of the handle ($p$), the position of the mass points representing the payloads relative to a target position ($x$), and their global velocities ($v$). The reward also contains a term that is a function of the actions which are taken. This term takes the form of a simple regularizer that aims to discourage large control actions.

$$r(s_t, a_t) \;=\; \frac{1}{\frac{1}{2}w_p||p_t||^2 + \frac{1}{2}w_x||x_t||^2 + \frac{1}{2}w_v||v||^2 + \frac{1}{2}w_a||a_t||^2}, \qquad (15)$$

where the coefficients $w_p, w_x, w_v, w_a$ allow each sub-objective to be weighted independently, as is commonly done. This very general reward formulation allows us to define two different tasks that we apply to each of the three systems described above:

- **Go to stop:** Starting from an initial state with non-zero velocity, the pendulum must go to stop as quickly as possible in a downward configuration. For this task the weights $w_p = w_x = 0$.
- **Go to stop at the origin:** In addition to stopping as fast as possible, the system must come to rest at a target location, which, without loss of generality, is chosen to be the origin.

The architecture of the neural network policies that we used is detailed in Appendix A.3. For a fair comparison, the neural network policies for PODS, PPO and SAC were initialized with the same set of initial weights. We fine tuned hyper parameters of PPO and SAC to get the best performance we could, and otherwise ran standard implementations provided in Achiam (2018).

## 4.1  RESULTS

The monotonically improving behaviour of PODS can be seen in Figure 5. The reward reported is the result of averaging the reward of 1000 rollouts started from a test bed of unseen initial states. Even if the initial progress of PODS is not always as fast as PPO or SAC, it consistently leads to a higher reward after a small number of epochs. We note that the standard deviations visualized in this figure are indicative of a large variation in problem difficulty for the different state-space points that seed the test rollouts (e.g. a double pendulum that has little momentum is easier to be brought to a stop than one that is swinging wildly). As can be seen, the tasks that demand the payloads to be brought to a stop at a specific location are considerably more challenging. The supplementary video illustrates the result of the rollouts to provide an intuition into the quality of the control policies learned with our method. Furthermore, Appendix A.6 presents convergence plots for the cable driven payload 2D, and the discretized 3D rope environments.

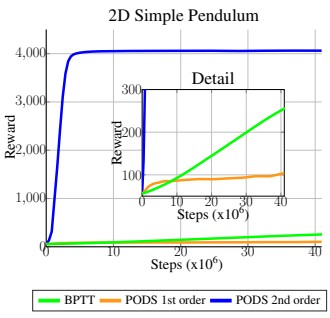

Figure 4: Comparison of PODS update rules against BPTT

**PODS vs BPTT:**  To further explore the benefits of the PODS second order update rule, we compared against the approach of BPTT which naturally leverages the differentiability of the model. We found BPTT to be highly sensitive to the weight initialization of the policy. In Figure 4, we report results using the weight initialization that we found to favor BPTT the most. When training neural network policies, doing BPTT for a 100 steps rollout is effectively equivalent to backpropagating through a network that is 100 times deeper than the actual network policy, which is in itself a feat considering that despite introducing a terminal cost function to stabilize BPPT, Clavera et al. (2020)

only reports results of effectively BPTT for a maximum of 10 steps. Nontheless, BPTT is able to outperform PODS with the 1st order update rule. However, PODS with the 2nd order update rule is able to significantly outperform BPTT both in terms on convergence rates and final performance. Even though, a second order formulation of BPTT could be derived, it's deployment would involve the hessian of the neural network policy which is computationally expensive. In contrast, PODS first order and second order formulations are equally easy to deploy.

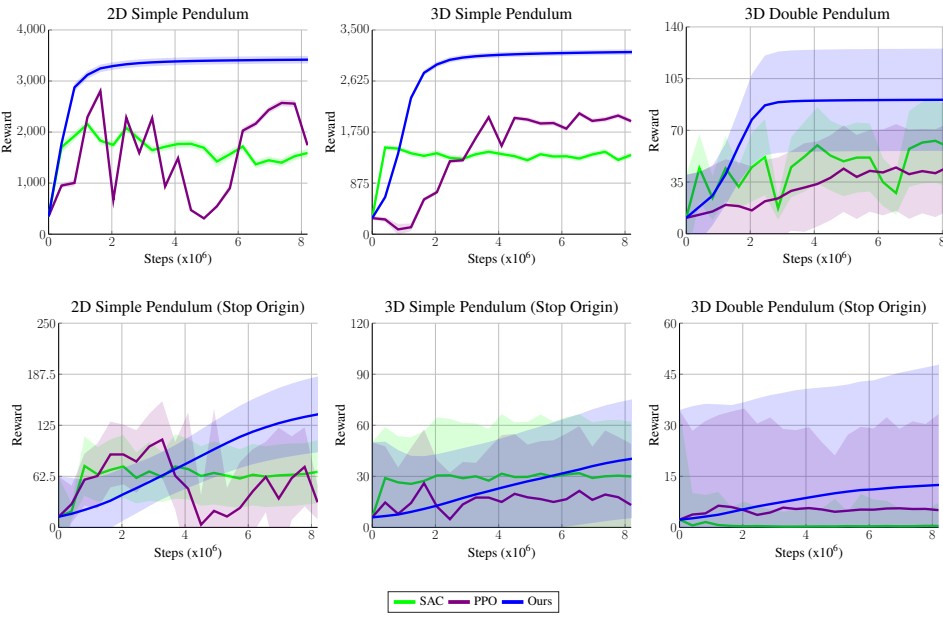

Figure 5: Comparison of reward curves. Our algorithm, PODS, achieves better performance compared to other algorithms, PPO and SAC

**PODS, SAC, and PPO:** To better understand the relative performance of the control policies learned with PODS, SAC and PPO, we report the terminal kinetic energy (KE) of the payload (Figure 6), the average magnitude of control action (Figure 8 – Appendix), and the average distance to the target location for the Stop At Origin tasks (Figure 7) – note, lower is better, and upon convergence, control policies learned with PODS adequately solve each individual problem in our randomized test bed. The shaded areas represent half the standard deviation of each metric. For figures with a logarithmic scale only the upper side of the standard deviation is presented.

For the task of stopping as fast as possible, PODS leads to a terminal kinetic energy that is typically orders of magnitude better than the other approaches (Top row Figure 6). For the tasks of stopping at the origin, SAC achieves very good terminal KE. The policies SAC learns, however, output large, high frequency handle motions, as seen in the high control cost in Figure 8. These actions end up countering the natural dynamic oscillations of the payload. The same strategy for the 3D double pendulum, however, is unsuccessful. In contrast, PODS learns control policies that use less effort to solve the control tasks than both SAC and PPO. This indicates that our policies learn to leverage the dynamics of the payload much more effectively, a characteristic that we attribute to the local improvement steps which, by design, monotonically improve the value functions of the control policies. Furthermore, it should also be noted that the class of fine manipulation tasks that we are dealing with represents a challenge for policies that output noisy actions.

## 5 CONCLUSION AND FUTURE WORK

In this paper, we presented a highly effective strategy for policy optimization. As a core idea behind our approach, we exploit differentiable simulators to directly compute the analytic gradient of a policy's value function with respect to the actions it outputs. Through specialized update rules,

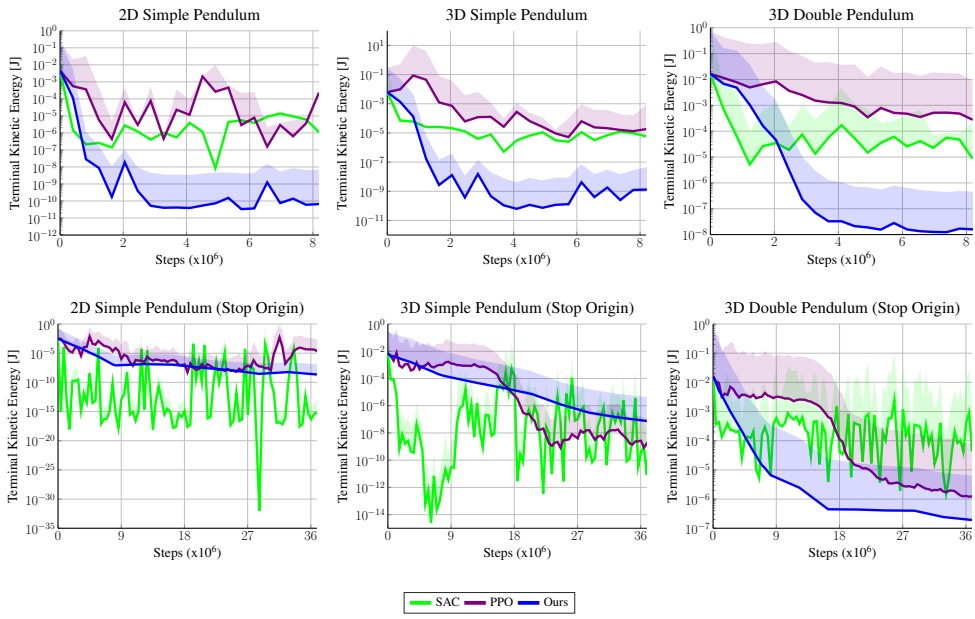

Figure 6: Final Kinetic Energy (averaged over a period of 10 time-steps after the policy is rolled out)

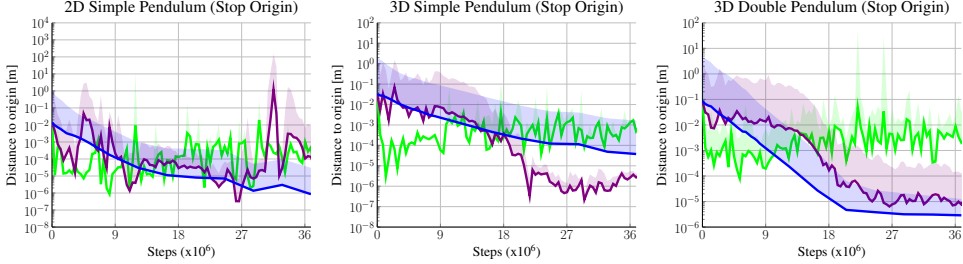

Figure 7: Final distance to Origin

this gradient information is used to monotonically improve the policy's value function. We demonstrated the efficacy of our approach by applying it to a series of increasingly challenging payload manipulation problems, and we showed that it outperforms two SOTA RL methods both in terms of convergence rates, and in terms of quality of the learned policies.

Our work opens up exciting avenues for future investigations. For example, although we evaluated PODS in isolation in order to best understand its strengths, it would be interesting to interleave it with existing RL methods. This will require extensions of our formulation to stochastic policies, and it would allow the relative strengths of different approaches to be effectively combined (e.g. exploration vs exploitation, with PODS excelling in the latter but not being designed for the former). We are also excited about the prospect of applying PODS to other types of control problems, particularly ones that include contacts (e.g. locomotion, grasping, etc). Although the need for a specialized simulator makes the application to standard RL benchmark suites (Brockman et al., 2016; Tassa et al., 2018) challenging, we note that sim-2-real success with a differentiable simulator has been recently reported in the context of soft locomoting robots (Bern et al., 2019). With continued evolution of such simulation technologies, we are excited about the prospect of creating a new benchmark suite applicable to approaches such as PODS that use differentiable simulators at their core.

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

## A APPENDIX

### A.1 VALUE FUNCTION HESSIAN

$$
\begin{aligned}
\frac{\mathrm{d}^2 V^{\bar{a}}(s_0)}{\mathrm{d}\bar{a}^2} &= \frac{\mathrm{d}}{\mathrm{d}\bar{a}} \left[ \frac{\partial V^{\bar{a}}}{\partial \bar{a}} + \frac{\partial V^{\bar{a}}}{\partial s} \frac{\mathrm{d}s}{\mathrm{d}\bar{a}} \right], \\
&= \frac{\mathrm{d}}{\mathrm{d}\bar{a}} \left[ \frac{\partial V^{\bar{a}}}{\partial \bar{a}} \right] + \frac{\mathrm{d}}{\mathrm{d}\bar{a}} \left[ \frac{\partial V^{\bar{a}}}{\partial s} \frac{\mathrm{d}s}{\mathrm{d}\bar{a}} \right], \\
&= \left[ \frac{\mathrm{d}s}{\mathrm{d}\bar{a}}^T \frac{\partial^2 V^{\bar{a}}}{\partial s \partial \bar{a}} + \frac{\partial^2 V^{\bar{a}}}{\partial \bar{a}^2} \right] + \frac{\mathrm{d}}{\mathrm{d}\bar{a}} \left[ \frac{\partial V^{\bar{a}}}{\partial s} \right] \frac{\mathrm{d}s}{\mathrm{d}\bar{a}} + \frac{\partial V^{\bar{a}}}{\partial s} \frac{\mathrm{d}}{\mathrm{d}\bar{a}} \left[ \frac{\mathrm{d}s}{\mathrm{d}\bar{a}} \right], \\
&= \left[ \frac{\mathrm{d}s}{\mathrm{d}\bar{a}}^T \frac{\partial^2 V^{\bar{a}}}{\partial s \partial \bar{a}} + \frac{\partial^2 V^{\bar{a}}}{\partial \bar{a}^2} \right] + \left[ \frac{\mathrm{d}s}{\mathrm{d}\bar{a}}^T \frac{\partial V^{\bar{a}}}{\partial s} + \frac{\partial^2 V^{\bar{a}}}{\partial \bar{a} \partial s} \right] \frac{\mathrm{d}s}{\mathrm{d}\bar{a}} + \frac{\partial V^{\bar{a}}}{\partial s} \left[ \frac{\mathrm{d}s}{\mathrm{d}\bar{a}}^T \frac{\partial}{\partial s} \frac{\mathrm{d}s}{\mathrm{d}\bar{a}} + \frac{\partial}{\partial \bar{a}} \frac{\mathrm{d}s}{\mathrm{d}\bar{a}} \right], \\
&= \frac{\partial V^{\bar{a}}}{\partial s} \left( \frac{\mathrm{d}s}{\mathrm{d}\bar{a}}^T \frac{\partial}{\partial s} \frac{\mathrm{d}s}{\mathrm{d}\bar{a}} + \frac{\partial}{\partial \bar{a}} \frac{\mathrm{d}s}{\mathrm{d}\bar{a}} \right) + \frac{\mathrm{d}s}{\mathrm{d}\bar{a}}^T \left( \frac{\partial^2 V^{\bar{a}}}{\partial s^2} \frac{\mathrm{d}s}{\mathrm{d}\bar{a}} + 2 \frac{\partial^2 V^{\bar{a}}}{\partial s \partial \bar{a}} \right) + \frac{\partial^2 V^{\bar{a}}}{\partial \bar{a}^2}.
\end{aligned}
$$

### A.2 DETAILED DESCRIPTION OF ENVIROMENTS

- **2D Simple Pendulum:** This system corresponds to a cable-driven pendulum in 2D (Figure 3 left). The handle of the pendulum is constrained to move only along the horizontal axis in order to test the degree to which a control policy can exploit the natural dynamics of the system.

- **3D Simple Pendulum:** For this system the pendulum is free to move in 3D, but the handle is restricted to moving along a horizontal plane.

- **3D Double Pendulum:** Extending the dynamical system above, the payload for this problem consists of two mass points that are coupled to each other via a stiff bilateral spring. The dimensionality of the state space doubles, and the system exhibits very rich and dynamic motions.

- **Cable drive payload 2D:** For this environment we have a densely connected network of 4 point masses and two handles that are constrained to move along the horizontal axis.

- **Rope in 3D:** For this environment we use 5 point masses to descretize a rope in 3D and one handle that is constrained to move on the horizontal plane.

Table 1: Summary of environments

| Environment | State space | Action space | Total mass | Constraints | Additional Info |
|---|---|---|---|---|---|
| 2D Simple Pendulum | 2 | 1 | 50gr | Handle along horizontal axis | |
| 3D Simple Pendulum | 3 | 2 | 50gr | Handle on horizontal plane | |
| 3D Double Pendulum | 6 | 2 | 100gr | Handle on horizontal plane | |
| Cable driven payload | 2*4 = 8 | 2 | 200gr | Handles along horizontal axis | Attachment to handle uses deformable cables |
| Rope in 3D | 3*5=15 | 2 | 250gr | Handle on horizontal plane | All conections are deformable cables |

### A.3 ARCHITECTURE OF NEURAL NETWORK POLICIES

The neural networks representing the control policies for all our environments share the same architecture, 2 fully connected layers of 256 units each with ReLU activations and one output layer with Tanh activation, to ensure that the policy only outputs commands that are within the velocity limits.

### A.4 PODS: ADDITIONAL FIGURES

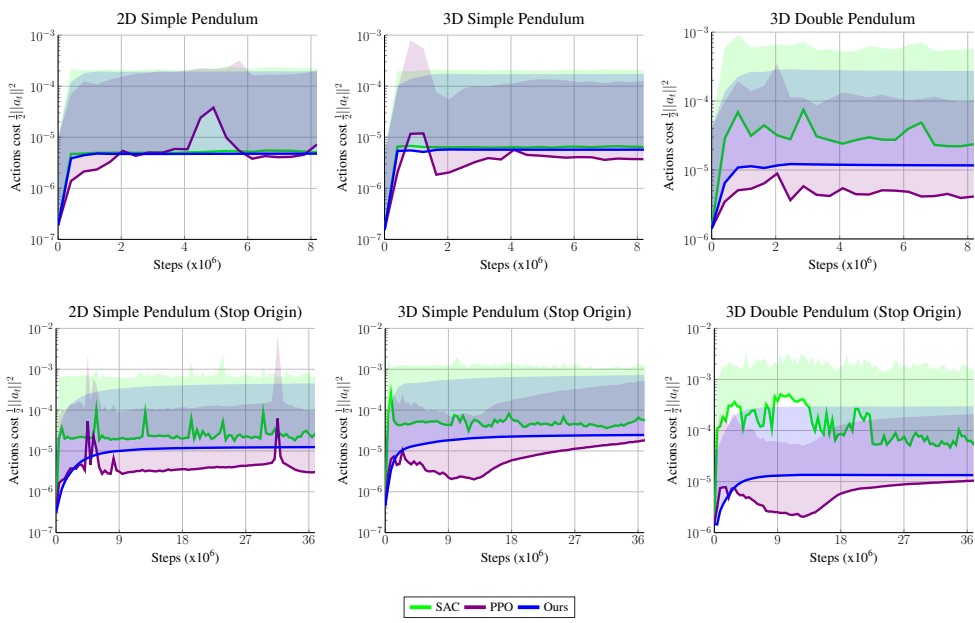

Figure 8: Average handle velocity (control effort)

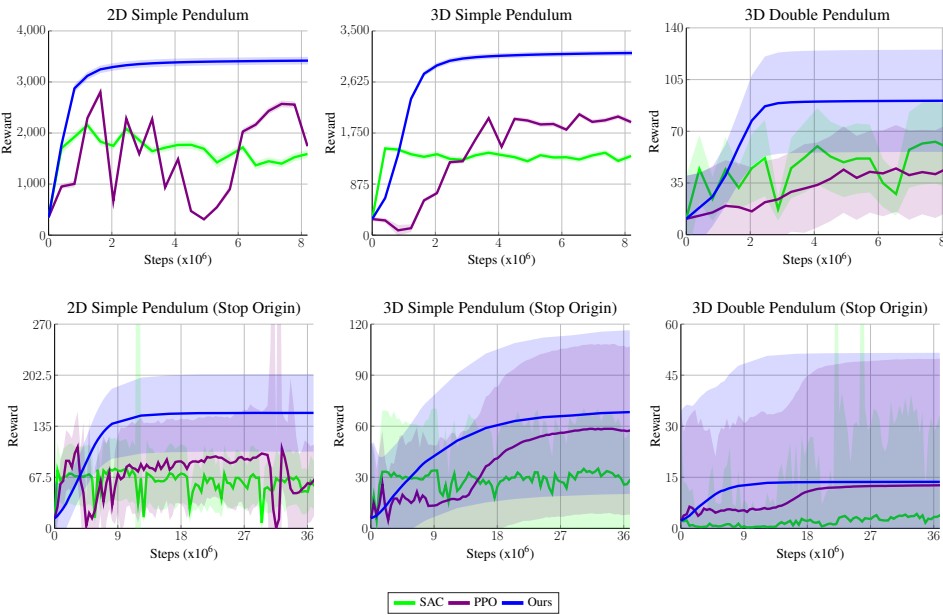

Figure 9: Extended Reward

## A.5 DIFFERANTIABLE SIMULATOR

Following the approach in Zimmermann et al. (2018), the sensitivity $\frac{\mathrm{d}s}{\mathrm{d}\bar{a}}$ has the structure of the figure below.

$$\frac{\mathrm{d}s}{\mathrm{d}\bar{a}} \quad = \quad -\left(\frac{\partial \mathbf{G}}{\partial s}\right)^{-1} \quad \frac{\mathrm{d}\mathbf{G}}{\mathrm{d}\bar{a}}.$$

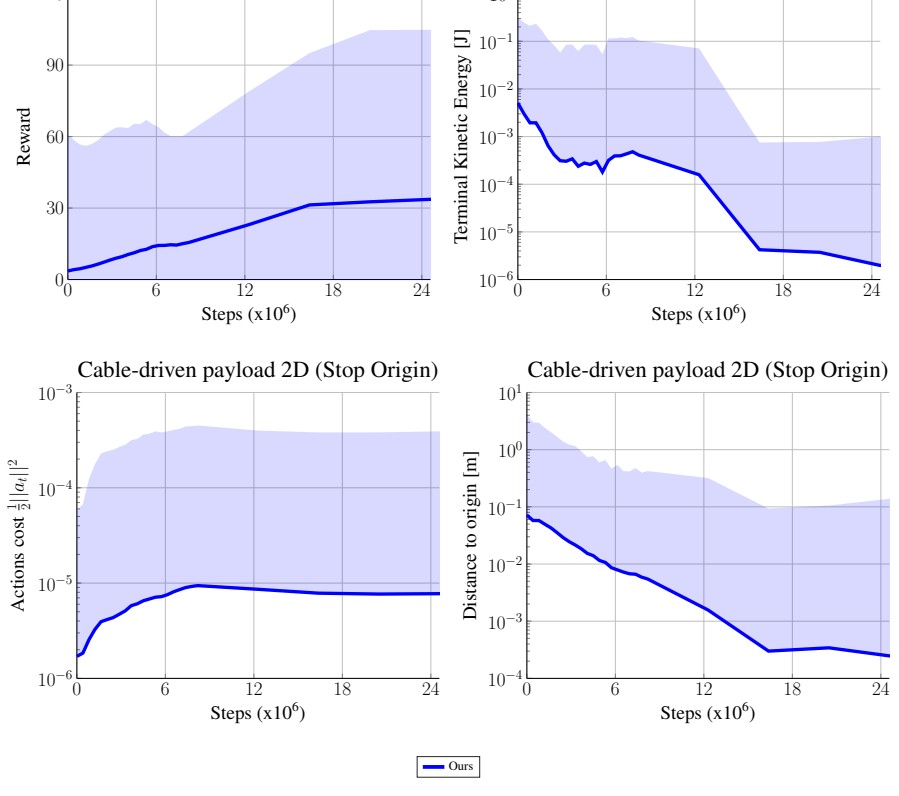

Figure 10: Structure of sensitivity matrix $\frac{\mathrm{d}s}{\mathrm{d}\bar{a}}$ that encodes the dependency of a state on all the previous actions [1]

## A.6 ADDITIONAL DEMOS

See the accompanying video for more details of the following environments.

Figure 11: Cable driven payload 2D

---

[1]Figure reproduced with authorization of the authors ( http://arxiv.org/abs/1905.08534 )

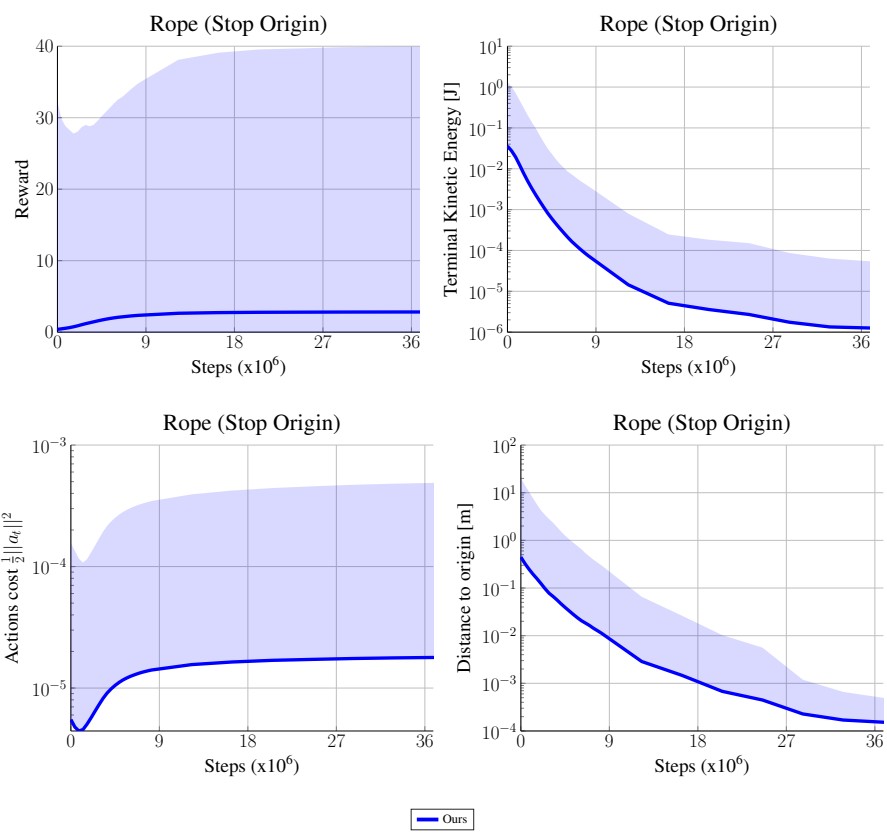

Figure 12: Rope in 3D

