# OpenReview forum: "PODS: Policy Optimization via Differentiable Simulation"
_ICLR.cc/2021/Conference — Reject_

### Official Review · AnonReviewer1 · 2020-10-28
**contribution of interest to the RL community**

**Rating:** 6
**Confidence:** 4

**Review:**

## Summary
The paper presents a class of RL algorithm based on analytic gradients of the objective,
coming directly from a fully differentiable simulator and differentiable rewards.
The analytic gradients are used to optimize trajectories.
The known states and actions from the optimized trajectories are then used to improve the deterministic policy.
Overall, this is an interesting paper that maximally leverages analytic gradient information
that come directly from a differentiable simulator, rather than a learned differentiable dynamics model.
The methods prove to be highly effective for the given class of problems they are tested on.

## Strengths
- This work provides strong insights into the benefits of differentiable simulators for
  trajectory-guided reinforcement learning.  The paper provides a solid discussion and evaluation of
  the use of first and second-order methods.
- It demonstrates effective and efficient results for a difficult class of problem that has direct application,
  i.e., cable-suspended loads

## Weaknesses
- The foundations of the trajectory-based optimization method exist in prior art [Zimmermann 2019]
- There is perhaps a restricted class of tasks can benefit from the approach:  contact collisions, infinite-horizons,
  and non-differentiable rewards may be off limits?

## Recommendation
The RL community can benefit from this work in several ways, as outlined with the various strengths given above.

## Questions
- Is the method restricted to fixed-length episodic tasks?  Fine if it is, but readers will want to know.
- Do local minima pose a problem, given that the method effectively does no true exploration?
- Can PODS be viewed as a form of fast analytic-derivative trajectory optimization followed by behavior cloning?
- Can hard constraints be handled, e.g., obstacles or joint limits that are to be avoided?
- What are the tradeoffs between working in handle-space vs robot joint space?

## Additional Feedback
What would be the impact of using many fewer rollouts per epoch?

Why not also compare the results for some tasks which are commonly used for RL, e.g., pendulum and acrobot swing-up?

Appendix A.2 only provides very minimal details on the environments.
What are the dimensions and masses of the various pendulum systems?

The following is an early paper (1998) that uses back-prop-through-time to directly optimize a control objective:
"NeuroAnimator:Fast Neural Network Emulation and Control of Physics-Based Models"

The proposed work has connections with Guided Policy Search (Levine and Koltun, IMCL 2013),
which also uses optimized trajectories as training data, only in their case for a stochastic policy
and using differential dynamic programming, instead of the deterministic policy setting used here
together with a Hessian-dependent Newton's method. Explaining these connections could be helpful
to many.

[ Rebuttal / question responses are acknowledged, and also the other reviews.
    I think the algorithm is sufficiently novel, and it does really well on some difficult new problems.
    I see the paper as being about a new policy optimization algorithm leveraging differential dynamics, and not about sim2real.
   However, the biggest limitation is pointed about by both R3 and R4, i.e., per R3: "With both algorithms, as well as the tasks, being new, its hand to establish the strength and credibility of both using one another. "  And so with this in mind, I am changing my score to a 6, i.e., marginally in favor of accept. ]

---

> ### Author Response · Authors · 2020-11-25
> **Response to Reviewer 1**
>
> ## Comments on weaknesses.
> - As progress in differentiable simulation is done to handle contact collisions, PODS could be used for such a class or problems. In particular we note that [1] proposes a differentiable framework that deals with rigid bodies and frictional contact.
>
> ADD: Analytically Differentiable Dynamics for Multi-Body Systems with Frictional Contact (Geillinger 2020) [1]
>
> - Regarding non differentiable rewards, we've started exploring the possibility of using a PODS-like strategy, leveraging both learned Q functions together with derivatives of a differentiable simulator,  in a finite horizon setting. Even if the reward it's not differentiable, the information that a differentiable learned Q function stores could be potentially useful for a policy optimization process.
>
> ## Answer 1:
>
> Currently the method is restricted to a finite horizon setting.
>
> ## Answer 2:
>
> We did not notice problems with poor local minima. This could be explained by the fact that PODS aims to simultaneously improve the performance for a large set of initial states and thus the resulting average policy update seems to be less prone to getting stuck in a poor local minima.
>
>
> ## Answer 3:
>
> See Answer 1 for Reviewer 4.
>
> ## Answer  4:
>
> By saturating the output using a tanh activation layer, some hard constraints can be satisfied. In our cases we limit our policies to output velocities of maximum +- 1.2 m/s. Additional constraints can be handled by introducing barrier functions in the reward.
>
> ## Answer 5:
>
> On one hand, handle space has lower dimensionality than robot joint space, which enables faster computation.  On the other hand, although robot joint space introduces further nonlinearities, planning in joint space allows to handle  joint limits more easily, thus ensuring that the motions are feasible and can be executed by the robot.
>
> This is also analogous to the difference between optimizing in trajectory space and parameter space for neural network policies.
>
> ## Comments on Feedback:
>
> - The number of rollouts per epoch iis something we would like to investigate further. A DAGGER strategy could certainly improve PODS performance. Moreover, we believe that the differentiable simulators could also be leveraged to come up with particular sets of points in state space, whose rollouts  contain more information and can be used to introduce other forms of curriculum learning.
>
> - More details on the dimensions and masses are now  presented in the appendix.
>
> - We appreciate the reference.
>
> - Relations to GPS is described in  Answer 1 to reviewer 4.

---

### Official Review · AnonReviewer4 · 2020-10-28
**A version of Guided Policy Search with differentiable simulators; limited novelty, suspicious evaluations**

**Rating:** 4
**Confidence:** 5

**Review:**

Summary
-------------
The paper argues for using differentiable simulators for policy optimization. To avoid back propagation through time, the paper splits the policy optimization problem into two steps: i) find improved action sequence for a set of initial conditions, ii) fit a parametric policy to the set of improved action sequences. First and second order methods are presented and evaluated on a version of payload-on-crane stabilization problem.

Decision
-----------
I vote for rejecting this paper for the following reasons.

1. There is little novelty in this work. The method can be seen as a version of Guided Policy Search (GPS) [1] where the trajectory optimization part is done using analytical gradients.
2. The evaluations don't seem to be reliable. In Fig. 5, PPO and SAC show very bad performance. Given that the task is quite low-dimensional, it is surprising that they work so badly. If that is nevertheless the case, then it would be beneficial to add experiments at least on the Classic Control environments from OpenAI Gym, such that one can compare to other papers.
3. The choice of the step size alpha is questionable. The paper says that unlike PPO and SAC, the proposed method does not require any policy update regularization. However, setting a small value of alpha effectively limits the policy update step. So, the proposed algorithm is still a version of conservative policy update algorithms, such as [1].

References
---------------
[1] Montgomery, W., & Levine, S. (2016). Guided policy search as approximate mirror descent. arXiv preprint arXiv:1607.04614.

---

> ### Author Response · Authors · 2020-11-25
> **Response to Reviewer 4.**
>
> We thank the reviewer for the insightful comments.
>
> ## Answer 1:
>
> Although PODS and GPS share a common goal, the mathematical formulation of both approaches is substantially different, as GPS is based on dual descent formulations while our approach is inspired by the policy gradient, which is also why we included the comparison against backpropagation through time.
>
> A first departure point of PODS w.r.t GPS is that at  each iteration the "control phase" or c-step reported in [1] requires to solve an optimization problem until convergence i.e. it requires a control oracle that internally performs several updates to the control actions, where each internal update also includes a line search procedure . PODS does not require such an oracle, as it only updates the control actions once per iteration by  simply following the gradient information from the differentiable simulation using a Newton method.
>
> If we think of updating the control actions once by following the gradient information as one of the many updates that a control oracle performs internally, then we can see that PODS is learning from the intermediate internal updates of an unregularized control oracle, while GPS only learns from the final solution of a regularized control oracle.
>
> In a sense PODS updates are more local than the ones in GPS. This makes our algorithm more on-policy and circumvents the need of KL divergence constraints, which GPS uses twice, once  in the control step (via a surrogate cost function) and a second time on the supervised step,  to make sure learning doesn’t collapse due to large changes of the policy and of the trajectory distribution, respectively.
>
> Furthermore, we note that the control phase of GPS uses iLQG or DDP algorithms which  compute locally optimal trajectories for a given initial state, while PODS computes policies that simultaneously maximize the reward for a large number of initial states. This is another advantage PODS has over GPS which suffers performance drops when the initial conditions do not belong to a narrow region of the state space [2]. This should come at no surprise since it is also mentioned by [1] that:
>
>     "Linear-Gaussian controllers represent individual trajectories with linear stabilization and Gaussian noise, and are convenient in domains where each local policy can be trained from a different (but consistent) initial state xi1 ∼ p(x1). This represents an additional assumption beyond standard RL, but allows for an extremely efficient and convenient local model-based RL algorithm"
>
> We acknowledge that a comparison against GPS could be useful and we have started working on it. However, implementing such a comparison has proven to be more challenging than initially expected. Which is why we would also like to highlight the simplicity of PODS w.r.t GPS as an additional advantage. Nonetheless, we expect to have the comparison ready within the next 2 weeks.
>
> Guided policy search as approximate Mirror Descent (Levine 2016) [1]
>
> Benchmarking Model-Based Reinforcement Learning (Wang et al. 2019) [2]
>
> ## Answer 2:
>
> We included a discussion of the relative performance of SAC, PPO, and PODS in the results section that was previously the appendix. We note that we use tuned standard available implementations of SAC and PPO. For the simplest tasks, SAC and PPO displayed a high frequency behavior that counteracts the dynamics of the systems to achieve the goal, as can be seen in the video in the supplementary material. Such high frequency behavior is reminiscent of locomotion controllers trained by SAC and PPO, where the agents behave well on average but exhibit jerky motions. Such strategy showed not to be successful as the dimensionality of the tasks increased. In a sense, these kind of fine manipulation tasks represent a torture test for algorithms that output noisy actions.
>
> ## Answer 3:
>
> Although we mentioned that $\alpha_a$ could be treated as a hyperparameter that should be set to a sufficiently small value, we used a standard backtracking line search in TS to adaptively set $\alpha_a$ for all of our experiments. For more details see Answer 4 to Reviewer 3.

---

### Official Review · AnonReviewer3 · 2020-10-28
**Paper needs better positioning, and appropriate supporting experimentation.**

**Rating:** 6
**Confidence:** 4

**Review:**

Pretext: Physics simulators are evolving and pose as a good approximation of the real world. Differentiable simulators are evolving and can provide analytic gradients.

Problem formulation: Can analytical gradients form differentiable simulators be leveraged to formulate better policy optimization algorithms?

Approach: Alternate between - optimization in trajectory space by explicitly taking gradients of the value function and - optimization in policy space by performing imitation learning over the improved actions.

Strengths: Tractable and stable optimization procedure due to the decoupling of the trajectory and policy optimization. Seconds order monotonic efficient method requiring minimal hyper parameter searches.

Discussions:
1. Inconsistencies between the positioning of the paper and experimental analysis
- Under the lens that the PODS primary contribution should be viewed as a new technique for policy optimization when environment-dynamics is differentiable, viewing SAC and POLO merely as a policy optimization algorithm (independent of training and test dynamics differences) is justified. Expectation under this viewpoint warrants comparison to similar (interleaving supervisedlearning with trajectory optimization) optimization paradigms like POLO(Plan online, learn offline: Efficient learning and exploration via model-based control
K Lowrey, A Rajeswaran, S Kakade, E Todorov), IGOR (Interactive control of diverse complex characters with neural networks. I Mordatch, K Lowrey, G Andrew, Z Popovic), GPS (Learning neural network policies with guided policy search under unknown dynamics S Levine, P Abbeel), etc.
- Under the premise that PODS' primary contribution should be viewed as a paradigm for sim2real, we need to account for the training and test (true) dynamics being different. While SAC and PPO should train using samples from true dynamics for training and report performance on true dynamics as well. PODS' should train only using samples from train-dynamics but report performance using true dynamics. Sample complexity should be compared only wrt to the sample drawn from the true dynamics (i.e. training sample for model-free methods and sample required for system-ID for PODS and similar methods)

2. No results on commonly accepted benchmarks tasks are presented. I recognize that the authors claimed that these tasks can pose for future benchmark tasks. But in this case, the focus and rigor for an evaluation will be completely different. With both algorithms, as well as the tasks, being new, its hand to establish the strength and credibility of both using one another. My suggestion will be to implement one/ two commonly accepted benchmark tasks and compare against the reported performance on them.

Suggestions:
1. Inline with point1 above, an interesting idea will be to perform an experiment in the spirit of sim2real. Where few established benchmarks tasks can be implemented with the differential engine. This is used as training-dynamics for training policies, and the resulting policy is evaluated for performance on equivalent OpenAI-Gym-mujoco as true dynamics.
2. "a" and "pi" are interchangeably called policies. Introducing a more crisp definition separating the two will help with clarity.
2. Missing figure reference in the appendix A2
3. Information on the line search for the experiments reported will be helpful.

---

> ### Author Response · Authors · 2020-11-25
> **Response to Reviewer 3**
>
> We thank the reviewer for the insightful comments.
>
> ## Answer 1:
>
> We are currently running parallel investigations regarding the sim-2-real transfer of PODS. We would like to point out that simulating the type of dynamic systems we are considering here requires implicit integration schemes. Stiff cables implemented as unilateral springs would demand exceedingly small time steps in a simulation framework that relies on explicit or semi-implicit integration, such as mujoco.  Furthermore, as mentioned in the conclusion we note that sim-2-real success is being increasingly showcased by differentiable simulators. The following link shows the result of one run of TO applied to the robot:
>
> https://drive.google.com/file/d/1sv9UaKEiYSh8mG8RfqCtI7GaKmJJNQct/view?usp=sharing
>
>
>  In this paper we focused on the mathematical formulation and experimental analysis of convergence, but early sim-2-real experiments are very promising and they will be thoroughly explored in a separate contribution.
>
> For a discussion regarding the relationship to GPS see Answer 1 to Reviewer 4.
>
> ## Answer 2:
>
> We changed the notation of $\boldsymbol{a}$ to $\boldsymbol{\bar{a}}$ to emphasize the fact that it is a policy in trajectory space. Furthermore, when introducing $\boldsymbol{\bar{a}}$ we are now explicitly noting that it is defined in trajectory space. We hope these changes will improve clarity.
>
> ## Answer 3:
>
> Reference was fixed.
>
> ## Answer 4:
>
> For all our experiments, we run a standard backtracking line search on the negative of the value function, starting from $\alpha_a=1$, which would be the right value if the value function was quadratic,and reducing it by halve at every iteration for a maximum of 50 iterations. We rarely encountered cases where the value function  $V^{\boldsymbol{\bar{a}}}$ was not improved after executing the line search.  Another advantage of defining $\boldsymbol{\bar{a}}$ is that line search in TS is feasible and efficient  as compared to line search in PS.
>
> The parameter $\alpha_\theta$, used for imitation learning, was set using an Adam optimizer. We note that satisfactory performance was also obtained using gradient descent, but ultimately Adam exhibited a better performance.

---

### Decision · Program_Chairs · 2021-01-07
**Final Decision**

**Decision:**

Reject

**Comment:**

The paper was evaluated by 3 knowledgeable reviewers, where 2 reviewers were leaning for acceptance and one reviewer argued for rejection, rendering the paper a borderline paper. The positive negative points that were raised about the paper during the discussion are summarized below:

Strength:
- The presented policy optimization method provides strong results
- It provided strong insights into the benefits of differentiable simulators for trajectory-guided reinforcement learning
- The direction of differentiable physics simulators is very promising and the provided benchmarks are interesting

Weak points:
- (i) The main contribution of the paper is a novel trajectory optimization method that uses analytical gradients. In a second step, a neural network is fitted to generalize the control from the single trajectories. The given approach is very much related to existing methods such as GPS or IGOR, just that the trajectory optimization is different. A comparison to these methods is needed. For example, how does the algorithm compare to using iLQG as trajectory optimization method ( we could also use analytical gradients for the linierazations used in iLQG)?
- (ii) While the presented tasks are very interesting, there is no benchmark on a more well known task. Hence, it is hard to evaluate the performance in comparison to other algorithms.

The paper defintely has interesting contributions in terms of the new trajectory optimization method and I could live with (ii) as the presented experiments are challenging and interesting, the contribution needs to be better evaluated as comparisions to other trajectory optimization methods are missing. I am sure that the paper will be accepted at another conference with this additional experiments, however, without it the paper is incomplete and I can unfortunately not recommend acceptance.